# The Effects of Alpha-Glycerylphosphorylcholine on Heart Rate Variability and Hemodynamic Variables Following Sprint Interval Exercise in Overweight and Obese Women

**DOI:** 10.3390/nu14193970

**Published:** 2022-09-24

**Authors:** Seyedeh Parya Barzanjeh, Linda S. Pescatello, Arturo Figueroa, Sajad Ahmadizad

**Affiliations:** 1Department of Biological Sciences in Sport, Faculty of Sports Sciences and Health, Shahid Beheshti University, Tehran 19839-69411, Iran; 2Department of Kinesiology, University of Connecticut, Storrs, CT 06269, USA; 3Department of Kinesiology and Sport Management, Texas Tech University, Lubbock, TX 79409, USA

**Keywords:** autonomic system, choline, blood pressure, exercise intensity, Wingate test

## Abstract

The current study examined the effects of Alpha-Glycerylphosphorylcholine (A-GPC) on heart rate variability (HRV) and hemodynamic responses following a sprint interval exercise (SIE) in women who were overweight or obese. Participants (*n* = 12, 31.0 ± 4.6 years; 29.4 ± 2.1 kg/m^2^) consumed 1000 mg of A-GPC or a placebo after eating breakfast in a randomized, double-blind cross-over design. After 60 min, participants performed two bouts of the SIE (30 s Wingate) interspersed with 4 min of active recovery (40 rpm). Hemodynamic variables and HRV domains were measured before and 60 min after the A-GPC consumption, immediately after SIE, and every 15 min up to 120 min during recovery. A-GPC consumption increased resting levels of both the time domain (Standard Deviation of RR wave intervals [SDNN] and percentage of interval differences of adjacent RR intervals greater than 50 ms [pNN50%]) and frequency domain (high frequency [HF] and low frequency [LF]) variables of HRV (*p* < 0.05). Moreover, HRV variables (except for LF/HF) decreased (*p* < 0.05) immediately after SIE in the A-GPC and placebo sessions. Systolic and diastolic blood pressure increased (*p* < 0.05) immediately after SIE in both trials. Both HRV and hemodynamic variables recovered (*p* < 0.05) faster in the A-GPC compared to the placebo session. We concluded that A-GPC consumption recovers HRV and blood pressure faster following strenuous exercise in overweight and obese women, and that it might favorably modify cardiac autonomic function.

## 1. Introduction

Cardiovascular diseases (CVDs) are the major cause of mortality and morbidity worldwide. Physical inactivity and obesity are major lifestyle CVD risk factors [1,2]. Heart rate variability (HRV) is a non-invasive method to evaluate cardiac autonomic control by measuring the time variation between consecutive RR intervals [3]. Previous studies have shown a relationship between cardiac autonomic function (measured by HRV) and obesity, which impairs HRV (decrease in high-frequency [HF] and increase in low-frequency [LF] power) [3,4]. Obesity-induced chronic inflammation and increases in reactive oxygen species (ROS) play an important role in sympathetic overactivity and reduction in vagal activity associated with visceral adiposity [4,5].

Vigorous exercise can lead to increased sympathetic activation and decreased parasympathetic activity, exaggerated blood pressure responses, acute cardiac events, and sudden cardiac death in susceptible individuals [6,7,8,9]. Sprint interval exercise (SIE), because of its time efficiency and physiological benefits, has attracted much attention in the last decade among both healthy people and patients with chronic diseases such as obesity [10,11,12]. However, SIE might acutely cause an enhanced risk of cardiac events during exercise and the recovery period through overactivation of the sympathetic nervous system (SNS) and impairment of cardiac function and hemodynamic responses [9,13]. Stuckey et al. (2012) compared cardiac autonomic and hemodynamic responses to single (one Wingate) and multiple sprint intervals (four bouts of Wingates) and found similar responses in HRV and hemodynamic variables; though, the recovery of HRV and hemodynamic variables took longer after four bouts compared to one bout [9]. Similarly, Ye et al. (2022) recently demonstrated that both two and four bouts of SIE (Wingate test) disturbed HRV and hemodynamic variables similarly in young sedentary individuals [13]. However, there is a dearth of information on the effects of SIE on cardiac autonomic function and hemodynamic variables during exercise and recovery in people with obesity.

Alpha Glycerylphosphorylcholine (A-GPC) is a food supplement that has gained growing attention in healthy individuals because of its favorable effects on the central nervous system (CNS) and possible protective influences on cardiovascular function [14,15,16,17,18]. After ingestion, it converts to phosphatidylcholine, the active form of choline that could enhance acetylcholine levels in the body [18,19]. Increased levels of free choline and acetylcholine may reduce ROS and inflammation, increase NO, and improve vagal activity [14,17,20,21]. Despite the possible cardioprotective effects of choline [21,22,23], its influence on cardiac autonomic function particularly following strenuous exercise that leads to surges in ROS, inflammation and disturbances of HRV and hemodynamic variables among adults with obesity is not clear.

Therefore, we hypothesized that the increased levels of free choline and acetylcholine that would result from consuming an A-GPC supplement could play a protective role on cardiac autonomic and blood pressure in people with obesity when they are exposed to a bout of SIE. Therefore, the present study was designed to investigate the effects of acute ingestion of A-GPC on HRV and blood pressure at rest and 2 h of recovery from SIE in overweight/obese individuals.

## 2. Material and Methods

### 2.1. Participants

Overweight or obese women (*n* = 12) were recruited through advertisements (posters, emails, and social media platforms). The characteristics of the participants are presented in Table 1. The inclusion criteria were: age between 20 to 40 years, body mass index (BMI) > 27 kg/m^2^, no regular exercise/physical activity, nonsmoker, no alcohol consumption in the last two months, and no diagnosed chronic diseases or health conditions. Participants were excluded if they used high choline diets, dietary supplements, or medication that could affect the subjects’ cardiac autonomic function and hemodynamic responses. All participants completed the physical questionnaire readiness (PAR-Q) [24] and medical health/history questionnaires. In addition, a written consent form was obtained prior to enrollment in the study. All participants were instructed to abstain from caffeine-containing beverages and any moderate to vigorous exercise/physical activity 24 h before reporting to the laboratory for testing. All study procedures were conducted according to the latest revision of the Declaration of Helsinki. The study procedures were reviewed and approved by the University’s Research and Ethics Committees (IR.SBU.REC.1400.228) and a clinical trial registry code was obtained (IRCT20160606028290N2).

### 2.2. Experimental Design

The first session was designed to familiarize participants with the Wingate test, A-GPC consumption, and other study procedures. In this session, height (Seca, HamburgGermany), body weight, body mass index (BMI), body fat percentage (BF%), and fat-free mass (FFM) were measured by the bio-impedance analyzer (X-Scan Inbody 770, Medigate Company Inc., Dan-dong Gunpo City, South Korea). In the second session, participants arrived at the laboratory in a fasting state (8 h) between 07:30 to 08:00 a.m. in the early follicular phase of menstruation (confirmed through calendar self-reporting method before participation). Participants consumed A-GPC (1000 mg) or a placebo after eating breakfast in a randomized, double-blind cross-over design with a one-week wash-out period between supplement and placebo trials. After consuming A-GPC or placebo, participants were seated for 60 min in a quiet room (temperature between 22 and 23 °C). Thereafter, they performed the SIE protocol. HRV was measured immediately after exercise and every 15 min up to 120 min during recovery. BP was measured immediately after exercise, every 5 min during the first 15 min of recovery, and afterwards every 15 min up to 120 min during recovery (Figure 1).

### 2.3. Heart Rate Variability

The participants arrived at the laboratory in the morning (07:30 to 08:00 a.m.). A heart rate monitor captured the RR recording (My patch-sl, 3-channles Holter, Los Angeles, CA, USA) in a seated position for 20 min before consuming A-GPC/placebo, 60 min after A-GPC/placebo consumption, and 120 min after the SIE bout. The first 5 min were ignored to establish signal stabilization and were not included in analyses. All the HRV measurements were conducted based on the standards proposed by the Task Force of the European Society of Cardiology [25]. A minimum protection zone of six beats was used to filtrate and correct all artifact and ectopic data. All recorded data with more than 5% error were excluded from the final analyses. Both the time and frequency HRV domains were analyzed using a computer software program (The Biomedical Signal and Medical Imaging Analysis Group, Department of Applied Physics, University of Kuopio, Kuopio, Finland) [26]. The time domain of HRV included the Standard Deviation of RR intervals (SDNN) and the percentage of interval differences of adjacent RR intervals greater than 50 ms (pNN50), which represent vagal modulation. The frequency domain parameters included the low-frequency (LF: 0.04–0.15 Hz) and high-frequency bands (HF: 0.15–0.40 Hz), which were calculated by using fast Fourier transformation (FFT; welch, 256 points Hanning-windowing, Kubios HRV Analysis, Biosignal Analysis, and Medical Imaging Group, University of Eastern Finland, Kuopio Finland) for analysis of the power spectrum. The ratio of the LF-to-HF (LF/HF) bands was also evaluated as an indicator of sympathovagal balance.

### 2.4. Blood Pressure Measurements

Systolic blood pressure (SBP), diastolic blood pressure (DBP), mean arterial pressure (MAP), and the rate pressure product (RPP) were evaluated before A-GPC/placebo consumption, 60 min after A-GPC/placebo consumption, and for 120 min after the Wingate test using an electronic sphygmomanometer (BPM AM 300P CE, Omron, Omron Company, Kyoto, Japan) based on guidelines of the European Society of Hypertension after A-GPC/placebo consumption [27]. Moreover, HR and SBP were used to calculate RPP (RPP = HR × SBP × 10).

### 2.5. A-GPC Supplementation

After consuming a standardized breakfast, participants consumed either 1000 mg of A-GPC (RAW powder United Kingdom) or placebo. Both the supplement and placebo were dissolved in 250 mL of water with a similar color and flavor. The A-GPC dose in this study was determined based on the study [15] conducted by Kawamura et al. (2012). According to the pharmacokinetics of the A-GPC supplement, it requires 60 min to reach its peak plasma levels and have maximal impact [28]. The University’s biochemistry laboratory analyzed and verified the supplement independently to determine its safety, presence of the active compound (99% l-alpha glycerylphosphorylcholine), and lack of any contaminant.

### 2.6. Sprint Interval Exercise (SIE)

In both the supplement and placebo trials the SIE (Wingate) protocol was performed at the same time of the day to control for diurnal variations. The protocol included a 5 min warm-up on an ergometer (Monark 894E Wingate testing bike ergometer, Sweden) at 40 to 60 rpm, which was followed by two 30 s SIE tests with 4 min active recovery (cycling at 40 rpm) between them. To determine the peak power, before starting the SIE test, participants were instructed to cycle as quickly as possible for 5 s, and then participants pedaled as fast as possible for 30 s encouraged verbally by a researcher. The load of 0.55 per kg body weight was placed on the fly-wheel and remained constant for 30 s to determine peak power (WPP) and mean power (WMP) in Watts.

### 2.7. Statistical Analyses

All data analyses were conducted with the Statistical Package for Social Sciences (SPSS) version 22 (IBM SPSS, Armonk, NY, USA). The Shapiro–Wilk test was used to determine that the data were normally distributed. Repeated measures ANOVA were employed to compare HRV indices (2 protocols × 11 times) and hemodynamics variables (2 protocols × 13 times) between the two trials. If significant differences were found, Bonferroni’s post-hoc comparisons were made. Independent t-tests were conducted to compare anaerobic power during the A-GPC and placebo trials. Effect size (ES) was determined by Partial eta-square, and power (1 − β) was assessed with repeated measures of ANOVA analyses. A *p*-value of *p* < 0.05 was considered statistically significant. All data are presented as mean ± SD.

## 3. Results

There were significant interactions (*p* < 0.05) between session and time for SDNN (F(2.9,32.4) = 11.2, *p* = 0.001, ES = 0.50, power = 0.99) (Figure 2A), pNN50% (F(2.5,28) = 9.2, *p* = 0.001, ES = 0.45, power = 0.98) (Figure 2B), HF (F(2.0,22.8) = 11.7, *p* = 0.001, ES = 0.51, power = 0.98) (Figure 3B), and LF (F(2.9,32.4) = 28.9, *p* = 0.001, ES = 0.72, power = 1.0) (Figure 3A). A-GPC consumption increased (*p* < 0.05) resting levels of SDNN (%14) (Figure 2A), pNN50% (%14) (Figure 2B), LF (%10) (Figure 3A), and HF (%13) (Figure 3B) compared to the placebo (Figure 2 and Figure 3). In both the A-GPC and placebo sessions these HRV variables decreased (*p* < 0.05) immediately after the SIE protocol, whereas, they increased during recovery to a greater level in the A-GPC than in the placebo session. There was a significant main effect of time for the LF/HF ratio (F(2.8,31.7) = 41.6, *p* = 0.001, ES = 0.79, power = 1.0) (Figure 3C). Independent of the trials (A-GPC or placebo), LF/HF (*p* < 0.05) increased immediately after the SIE protocol and decreased during 120 min recovery. However, there were no significant interactions between session and time for the LF/HF ratio (*p* > 0.05) (Figure 3C). 

There were significant interactions between session and time for SBP (F(3.3,37.2) = 23.5, *p* = 0.0001, ES = 0.68, power = 1.0) (Figure 4A), HR (F(2.8,31.3) = 33.2, *p* = 0.0001, ES = 0.75, power = 1.0) (Figure 5A), and RPP (F(2.6,29.4) = 44.4, *p* = 0.0001, ES = 0.80, power = 1.0) (Figure 5B). Although, SBP, HR and RPP increased (*p* < 0.05) immediately after the SIE protocol in both trials, SBP increased to a lesser level (*p* < 0.05) in the A-GPC (%47) than in the placebo (%64) trial (Figure 4 and Figure 5). SBP, HR and RPP decreased (*p* < 0.05) during recovery in both trials, though they recovered (*p* < 0.05) faster in the A-GPC compared to the placebo session (Figure 4A). Although, a significant main effect of time was observed for DBP (F(2.9,32.2) = 11.5, *p* = 0.0001, ES = 0.51, power = 0.99), DBP (*p* < 0.05) increased immediately after the SIE protocol and decreased during recovery with no significant difference (*p* > 0.05) between the two trials (Figure 4B).

Peak and mean power were (*p* < 0.05) higher in the A-GPC compared to the placebo session (Table 2).

## 4. Discussion

A-GPC is a food or health supplement that converts to phosphatidylcholine, the active form of choline after ingestion, increasing acetylcholine and free plasma choline [17,18]. However, its influence on cardiac autonomic function, particularly following strenuous exercise, is unclear among adults who are overweight or obese. The main findings of the present study were: (1) A-GPC consumption alone improved both time and frequency domains of HRV variables, (2) these HRV indices decreased and hemodynamic variables increased more pronouncedly immediately after SIE in both A-GPC and placebo trials, which show disruption effects of SIE on cardioautonomic function and hemodynamic variables, and (3) the recovery of HRV (SDNN, pNN50%, HF and LF) and hemodynamic (SBP, HR and RPP) variables were faster in the A-GPC trial, representing the effectiveness of A-GPC in the modulation of both cardioautonomic and hemodynamic variables following SIE in overweight/obese women. Therefore, our initial hypothesis regarding the proper effects of A-GPC on the responses of HRV and hemodynamic indices to SIE was confirmed. Although A-GPC after ingestion might increase acetylcholine and free plasma choline [14,15,17], the precise mechanisms of choline effects on ANS are not clear. Liu et al. (2017) indicated that short-term choline supplementation in hypertensive rats was associated with improvement of vagal activity through its effective role in increasing baroreceptors sensitivity [21]. Moreover, they showed that choline supplementation could have cardioprotective effects by inhibiting toll-like receptor 4, pro-inflammatory cytokines, and upregulation of anti-inflammatory markers [21]. Therefore, although in our study, we did not measure inflammatory markers, the more pronounced favorable changes in HRV and hemodynamic variables following A-GPC consumption, in the present study, might be justified by its role in the inhibition of inflammation and oxidative stress [14,17,29,30]. Another possible mechanism that might be used to justify these findings is the effect of choline on NO stimulation that might modify the cardiac autonomic system (increase HRV) and hemodynamic variables (reduced SBP, DBP and RPP) through reactivation of vagal activity and its vasodilation demeanor [14,20,31]. Since we did not measure NO, oxidative stress and inflammatory markers, more studies are required to shed more light on the mechanisms responsible for ANS and hemodynamic responses to exercise following choline consumption. In general, we found that A-GPC could have a cardioprotective effect following SIE, and these findings would suggest the clinical implication of A-GPC consumption in reducing the risk of cardiac events in overweight/obese women following vigorous exercise.

In our study, the SIE protocol reduced HRV variables while increasing hemodynamic variables, which confirms our original hypothesis and the findings of previous studies [13]. For example, Stuckey et al. (2011) and Ye et al. (2022), demonstrated that SIE protocols (Wingate base) were associated with imposing significant stress on ANS and hemodynamic variables, which might lead to increased risk of cardiac events after these supramaximal exercises [9,13]. Previous studies have shown that vigorous exercise is associated with stimulation of sympathetic activation leading to increased catecholamine production [32,33]. Furthermore, this supramaximal intensity exercise imposes substantial stress, which can cause muscle damage, increase oxidative stress and inflammatory markers and consequently sympathetic overstimulation and delayed parasympathetic reactivation during both exercise and recovery [30,34,35].

Another interesting finding of our study is that acute ingestion of 1000 mL/g of A-GPC improves the peak power, mean power, and fatigue index during the Wingate test. Our results confirm the finding of others who demonstrated increases in mean power, power, and isometric strength during a maximal effort sprint test [36,37]. A-GPC-induced improvements in performance are possibly related to its potential role in increasing choline bioavailability and consequently acetylcholine synthesis augmentation levels in CNS neurons [38]. Moreover, it was shown that A-GPC could improve performance by increasing the growth hormone concentration [15]. However, future studies are warranted to confirm the ergogenic effects of A-GPC on anaerobic performance.

A limitation of the present study was that HRV mostly represents parasympathetic activity and to determine the sympathetic activity, we suggest future studies employ other techniques such as microneurography and systolic time intervals. Moreover, this study focused on the acute consumption of A-GPC, while chronic effects of A-GPC consumption and exercise training might lead to important and solid findings for human health. Therefore, future studies are warranted to shed some more light on the chronic impact of A-GPC on autonomic and hemodynamic variables. Another limitation was the low number of young healthy participants. Thus, our findings may not apply to middle-aged and older adults.

The strength of the current study is that it was performed in a double-blind, randomized cross-over design, which makes the study results more valuable. Additionally, our study shows the clinical implication of A-GPC consumption prior to vigorous exercises by modulation of cardiac autonomic and hemodynamic responses to exercise and recovery, which might be useful for obese individuals for preventing cardiac events and encouraging better recovery. However, more studies are needed to confirm these results and their practical application, particularly by addressing the chronic effects of A-GPC on ANS.

## 5. Conclusions

We found the A-GPC consumption prior to SIE can attenuate the changes in HRV and hemodynamic variables that typically occur during and after SIE and recovery. Therefore, it may be a useful strategy to reduce the risk associated with supramaximal exercise in individuals who are overweight or obese. Additionally, our study indicates that A-GPC could have ergogenic effects on performance by improving peak power, mean power, and fatigue index. However, more studies are required to determine the mechanisms responsible for A-GPC-induced modulations in ANS and hemodynamics following supramaximal exercise.

## Figures and Tables

**Figure 1 nutrients-14-03970-f001:**
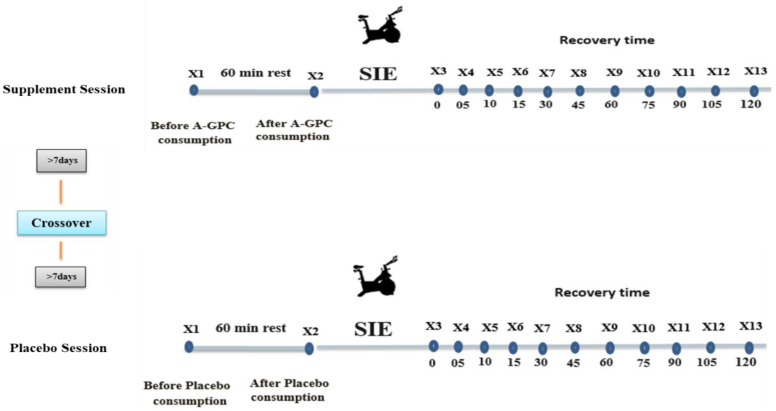
Research design (X = Measurement).

**Figure 2 nutrients-14-03970-f002:**
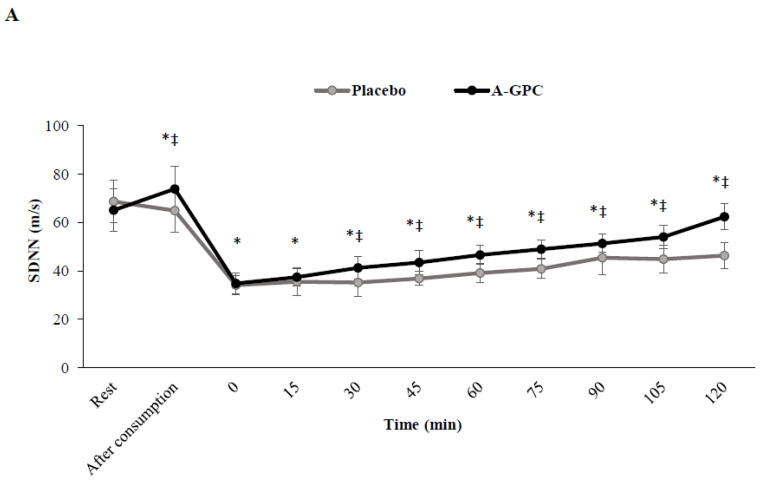
SDNN (**A**) and PNN50% (**B**) at baseline, 60 min after choline or placebo consumption immediately after exercise and every 15 min after exercise. SDNN, Standard Deviation of RR wave intervals; pNN50%, percentage of interval differences of adjacent RR intervals greater than 50 ms. Values are means ± SD. * indicates a significant difference from baseline (*p* < 0.05); ‡ indicates a significant difference from placebo (*p* < 0.05).

**Figure 3 nutrients-14-03970-f003:**
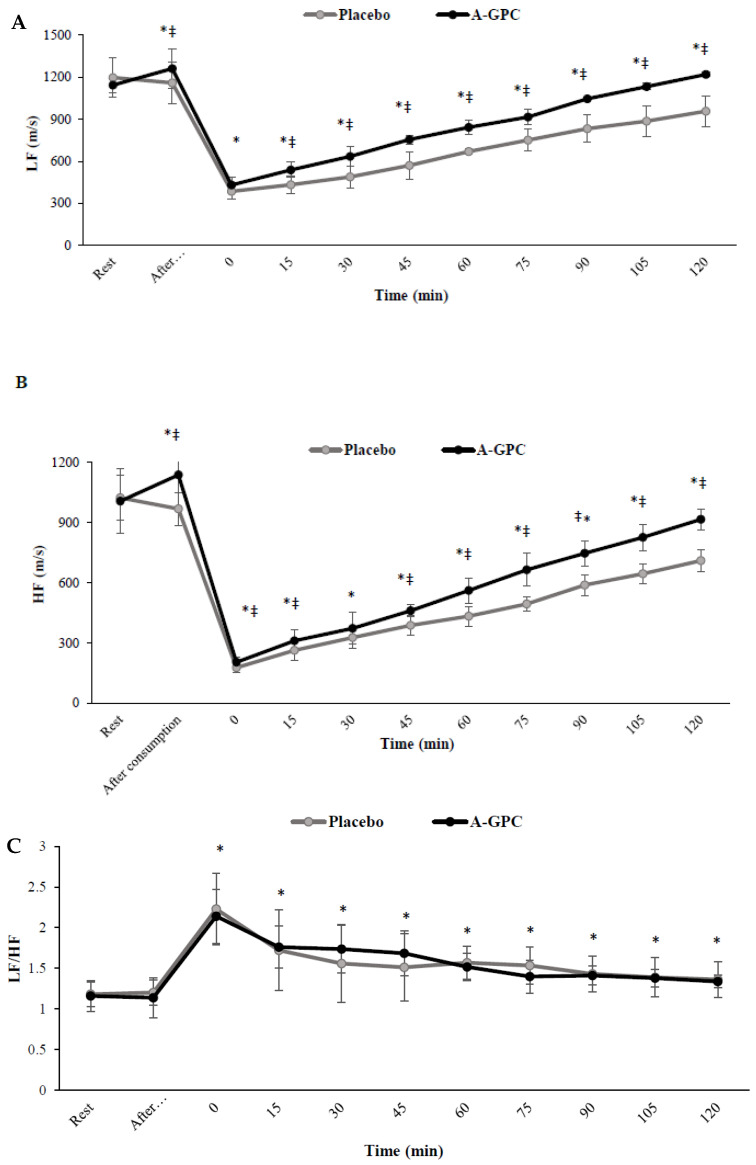
Low-frequency (**A**), high-frequency (**B**) and low-frequency/high-frequency ratio (**C**) at baseline, 60 min after choline or placebo consumption, immediately after exercise and every 15 min after exercise. LF, low frequency; HF, high frequency. Values are means ± SD. * indicates a significant difference from baseline (*p* < 0.05); ‡ indicates a significant difference from placebo session (*p* < 0.05).

**Figure 4 nutrients-14-03970-f004:**
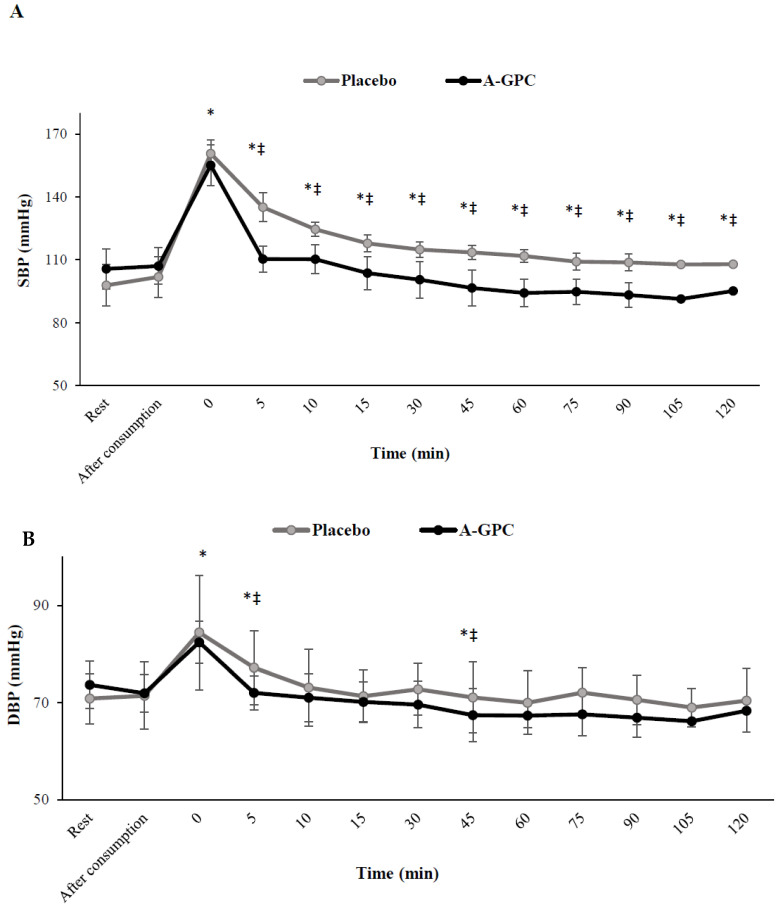
Systolic (**A**) and diastolic (**B**) blood pressure at baseline, 60 min after Choline or placebo consumption immediately after exercise and every 5 min during first 15 min and afterward every 15 min. SBP, Systolic Blood Pressure; DBP, Diastolic Blood Pressure. Values are means ± SD. * indicates a significant difference from baseline (*p* < 0.05); ‡ indicates a significant difference from placebo session (*p* < 0.05).

**Figure 5 nutrients-14-03970-f005:**
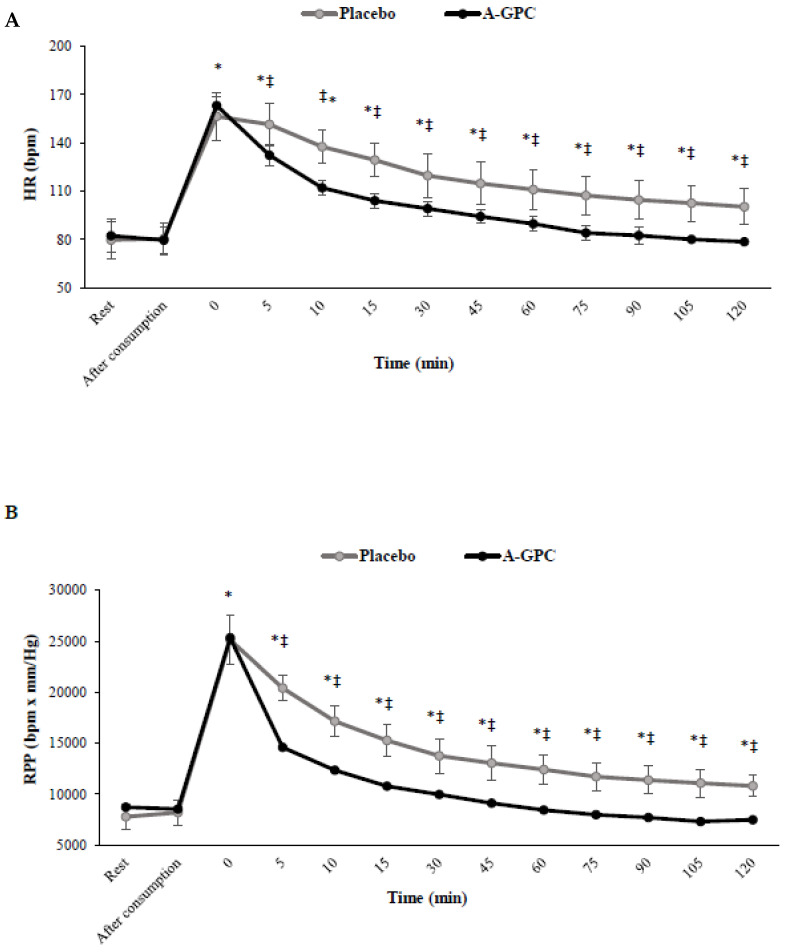
Heart rate (**A**) and Rate Pressure Product (**B**) at baseline, 60 min after choline or placebo consumption immediately after exercise and every 5 min during first 15 min and afterward every 15 min. HR, Heart Rate; RPP, Rate Pressure Product. Values are means ± SD. * indicates a significant difference from baseline (*p* < 0.05); ‡ indicates a significant difference from placebo session (*p* < 0.05).

**Table 1 nutrients-14-03970-t001:** Participant Baseline Characteristics (*n* = 12).

Variable	Means ± SD
Age (year)	31.0 ± 4.6
Height (cm)	160 ± 7.2
Weight (kg)	75.3 ± 4.0
BMI (Kg/m^2^)	29.4 ± 2.1
BF (%)	37.7 ± 4.8
SDNN (m/s)	66.7 ± 8.8
pNN50 (%)	30.5 ± 6.3
LF (m/s)	1168 ± 113
HF (m/s)	1091 ± 158
LF/HF (ratio)	0 ± 0.1
SBP (mmHg)	109 ± 5.8
DBP(mmHg)	73.2 ± 6.8
HR (bpm)	76.7 ± 9.2
RPP (bpm×mmHg)	8360 ± 921

BMI, body mass index; BF, body fat percent; SDNN, Standard Deviation of RR wave intervals; pNN50%, percentage of interval differences of adjacent RR intervals greater than 50 ms; LF low frequency; HF, high frequency; SBP, systolic blood pressure; DBP, diastolic blood pressure; HR, heart rate; RPP, rate pressure product.

**Table 2 nutrients-14-03970-t002:** Mean (±SD) values of power indices (*N* = 12).

Characteristics	Placebo 1st Bout	Supplement 1st Bout	Placebo 2nd Bout	Supplement 2nd Bout
Peak power (W)	407 ± 63.1	465 ± 88.1 ^‡^	362 ± 54.5	416 ± 77.4 ^‡^
Mean power (W)	304 ± 40.4	338 ± 59.6 ^‡^	257 ± 45.6	292 ± 69.9 ^‡^
Fatigue index (%)	55.8 ± 15.8	51.8 ± 6.0	60.4 ± 10.2	57.7 ± 9.1

W, watt. ^‡^ indicates a significant difference from placebo.

## Data Availability

The datasets generated for this study are available on request to the corresponding author.

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
