# Peer review of "The Effects of Alpha-Glycerylphosphorylcholine on Heart Rate Variability and Hemodynamic Variables Following Sprint Interval Exercise in Overweight and Obese Women"

_nutrients, 2022, doi:10.3390/nu14193970_

Round 1

Reviewer 1 Report

3. You can use “training” instead of “exercise”. You can find it in the literature as SIT. Moreover, it's an acute effect study. Thus, this must be shown in the title.

13. Avoid 1st plural. Please, check the whole text.

15. It is not necessary again to refer to overweight to obese women.

16. Delete “after eating breakfast”.

18. Refer to some aspects of SIT. Duration, intensity, volume, etc.

20-22. Please explain the abbreviations. 

36. Can you give some extra information about the impairment of HRV?

36-40. Please check again the sentence.

43. I don’t think that is necessary to use “supramaximal” before “sprint”.

57. It is better to split the introduction into two parts. One for SIT and another for Alpha Glycerylphosphorylcholine.

70. Please, write clearer the purpose of your study.

80. Delete “.” inside the parentheses.

90 Add under the table analytical the abbreviations. Also, the table must show information that has been already shown in the text. Thus, it is better to move it below the 2.2 section.

105-107. Move this sentence to the 2.1 section.

111.-112. Delete this description. It was used before. Say “at the same place” or something like that.

119. How many participants were excluded? This exclusion affected the final number of participants?

159. Was the distribution of your sample normal? 

172. Write the results in that way (F(2.9, 32.4) = 11.2, p = 0.001).

227. It is better to use past tense.

291. Write separately the conclusion, adding the title of “conclusions”.

Author Response

Thank you for reviewing our manuscript, which has been revised following your comments.

Reviewer 2 Report

Congratulations to the authors for the work presented.

The introduction is appropriate and pertinent to the study carried out

The methodology is consistent and very well developed.

The analysis for the treatment of the data is adequate

Author Response

We appreciate your comments.

Thank you,

Reviewer 3 Report

In my opinion, the manuscript needs revisions. Therefore, I have included my comments below: 

The Introduction Section explains the design of the study. 

The Material and Method section - Please explain the recruitment of such a small number of people for the study. The work carried out makes it possible to increase the number of subjects to one that will give a more substantial inference. 

The Descriptions of the results were correct. 

The presented figures and table were prepared precisely and also legible. 

The Discussion Section included the accurate reference of the results obtained from the studies of other authors. However, I believe that the number of respondents is a limitation of the study.  

The Conclusions were well formulated.

Author Response

We appreciate your carful revision of the manuscript. Our responses to your comments are in the attached file. Thank you.

Round 2

Reviewer 1 Report

38. Check the sentence grammatically.

299. Avoid 1st plural.

Generally, check for grammatical errors.